# The Renin–Angiotensin System Modulates SARS-CoV-2 Entry via ACE2 Receptor

**DOI:** 10.3390/v17071014

**Published:** 2025-07-19

**Authors:** Sophia Gagliardi, Tristan Hotchkin, Hasset Tibebe, Grace Hillmer, Dacia Marquez, Coco Izumi, Jason Chang, Alexander Diggs, Jiro Ezaki, Yuichiro J. Suzuki, Taisuke Izumi

**Affiliations:** 1Department of Biology, College of Arts & Sciences, American University, Washington, DC 20016, USA; sg1978a@american.edu (S.G.); th9716a@american.edu (T.H.); ht8146a@american.edu (H.T.); gh2297a@american.edu (G.H.); dm6732a@american.edu (D.M.); cizumi@american.edu (C.I.); jchang@american.edu (J.C.); ad6045a@american.edu (A.D.); jezaki@american.edu (J.E.); 2Department of Pharmacology and Physiology, Georgetown University Medical Center, Washington, DC 20007, USA; ys82@georgetown.edu; 3District of Columbia Center for AIDS Research, Washington, DC 20052, USA

**Keywords:** angiotensin, renin–angiotensin system, angiotensin-converting enzyme 2, severe acute respiratory syndrome coronavirus 2, spike protein, viral-like particle

## Abstract

The renin–angiotensin system (RAS) plays a central role in cardiovascular regulation and has gained prominence in the pathogenesis of Coronavirus Disease 2019 (COVID-19) due to the critical function of angiotensin-converting enzyme 2 (ACE2) as the entry receptor for severe acute respiratory syndrome coronavirus 2 (SARS-CoV-2). Angiotensin IV, but not angiotensin II, has recently been reported to enhance the binding between the viral spike protein and ACE2. To investigate the virological significance of this effect, we developed a single-round infection assay using SARS-CoV-2 viral-like particles expressing the spike protein. Our results demonstrate that while angiotensin II does not affect viral infectivity across concentrations ranging from 40 nM to 400 nM, angiotensin IV enhances viral entry at a low concentration but exhibits dose-dependent inhibition at higher concentrations. These findings highlight the unique dual role of angiotensin IV in modulating SARS-CoV-2 entry. In silico molecular docking simulations indicate that angiotensin IV was predicted to associate with the S1 domain near the receptor-binding domain in the open spike conformation. Given that reported plasma concentrations of angiotensin IV range widely from 17 pM to 81 nM, these levels may be sufficient to promote, rather than inhibit, SARS-CoV-2 infection. This study identifies a novel link between RAS-derived peptides and SARS-CoV-2 infectivity, offering new insights into COVID-19 pathophysiology and informing potential therapeutic strategies.

## 1. Introduction

Severe Acute Respiratory Syndrome Coronavirus 2 (SARS-CoV-2), the causative agent of Coronavirus Disease 2019 (COVID-19), is an enveloped, positive-sense, single-stranded RNA virus belonging to the Betacoronavirus genus [1,2]. Its genome encodes four major structural proteins, nucleocapsid (N), membrane (M), envelope (E), and spike (S), as well as 16 non-structural proteins involved in viral replication and host interaction [3]. Among these, the spike glycoprotein plays a pivotal role in viral entry by mediating host cell recognition and membrane fusion [4]. This protein comprises two functional subunits: S1, which contains the receptor-binding domain (RBD) responsible for engaging host receptors, and S2, which facilitates viral and host membrane fusion. The primary receptor for SARS-CoV-2 entry is angiotensin-converting enzyme 2 (ACE2), a surface carboxypeptidase expressed in various human tissues [5,6,7]. Upon binding of the RBD spike protein to ACE2, the spike protein undergoes proteolytic activation by host proteases such as TMPRSS2, triggering conformational changes that allow the virus to penetrate the host cell [5,6,8]. In the absence or low expression of TMPRSS2, SARS-CoV-2 can instead utilize an endocytic pathway, where the spike protein is cleaved by cathepsin L within the acidic environment of endosomes to enable membrane fusion and entry [9]. Despite the central role of ACE2 receptor in viral entry, its expression is variable across tissues, and relatively low in the lungs, prompting ongoing investigation into factors that modulate viral access and entry efficiency [10]. ACE2 is also a critical regulator of the renin–angiotensin system (RAS), a hormonal cascade essential for cardiovascular and renal homeostasis [11,12]. Within this system, angiotensinogen is cleaved by renin to form angiotensin I, which is subsequently converted to angiotensin II by ACE (Figure 1). Angiotensin II primarily signals through the angiotensin II type 1 receptor (AT1R), eliciting vasoconstriction, pro-inflammatory responses, and fluid retention [13]. However, angiotensin II can be further processed by aminopeptidases to generate downstream metabolites with distinct biological roles.

Aminopeptidase A (APA) cleaves angiotensin II to produce angiotensin III, which retains affinity for both AT1R and the angiotensin II type 2 receptor (AT2R), but with reduced potency [14]. Angiotensin III is further degraded by aminopeptidase N (APN) into angiotensin IV, a peptide that exerts effects through the AT4 receptor (AT4R), also known as insulin-regulated aminopeptidase (IRAP) [15,16]. AT4R is implicated in diverse physiological functions, including natriuresis, vasodilation, and cognitive processes such as learning and memory [17,18]. Given the dual role of ACE2 in RAS regulation and as the SARS-CoV-2 entry receptor, interactions between angiotensin peptides and viral entry mechanisms are of growing interest. Recent studies have suggested a complex interplay between the hormonal milieu of the RAS and SARS-CoV-2 infection. The spike protein leads to ACE2 downregulation via interaction, which contributes to the loss of ACE2-induced protective effects and stimulation of counter-regulatory angiotensin II-induced pathogenesis, promoting vasoconstriction, inflammation, and oxidative stress, thereby exacerbating COVID-19 pathogenesis [19]. Mathematical models have suggested that SARS-CoV-2-induced ACE2 downregulation perturbs RAS homeostasis, and interventions targeting RAS components predicted an improvement of the clinical outcome of COVID-19 for some drugs and a worsening for others [20]. This indicates that SARS-CoV-2 infection also influences the balance of the RAS cascade, which leads to severe COVID-19 diseases. It has been reported that ACE2 expression is higher in females than in males, leading to increased circulating levels of angiotensin (1–7) in females [21,22], while angiotensin II tends to be higher in males [23,24,25]. When comparing COVID-19 outcomes between males and females, a clear trend emerges that males are hospitalized more frequently and are more likely to require intensive interventions such as intubation and vasopressor support [26,27]. Furthermore, hospitalized male patients face a higher risk of thrombosis and mortality compared to their female counterparts [28]. On the other hand, females exhibit a higher incidence of neurological symptoms associated with COVID-19, such as fatigue, headache, brain fog, depression, and anosmia, compared to males [29]. This suggests that the SARS-CoV-2 brain reservoir, particularly within microglial cells [30] that drive neuroinflammation, may be larger or more active in females.

The underlying causes of these disparities remain unclear, but they may be attributed to various biological differences between the sexes, and one potential factor may be the differences in circulating angiotensin peptides and their biological activities.

Recent findings suggest that angiotensin peptides may influence not only COVID-19 complications but also viral infectivity. We recently reported that ELISA-based spike protein binding assays demonstrated that, while angiotensin II has no significant effect, its smaller downstream metabolite angiotensin IV enhances the binding affinity of the SARS-CoV-2 spike protein to the ACE2 receptor in vitro [31,32]. To further characterize the virological effect of angiotensin IV, we investigated its impact on SARS-CoV-2 infectivity and found that enhanced spike–ACE2 interaction correlates with increased viral entry at physiologically relevant concentrations of angiotensin IV. The greater stability and abundance of angiotensin IV in the central nervous system (CNS) suggests that it may predominantly contribute to the neurological complications associated with COVID-19. Angiotensin IV-mediated enhancement of SARS-CoV-2 cell entry could increase the viral reservoir in the brain, particularly in females, potentially contributing to the higher prevalence of COVID-19–related neurological symptoms in women.

## 2. Materials and Methods

### 2.1. In Silico Docking Simulation

Protein–ligand docking simulations were conducted to evaluate interactions between angiotensin peptides and SARS-CoV-2 spike protein structures. The crystal structures used were the RBD of the SARS-CoV-2 spike protein bound to ACE2 (PDB ID: 6M0J) [33] and the trimeric closed and opened forms of the spike glycoprotein (PDB ID: 6VXX and 6VYB, respectively) [34]. Simulations were performed using DynamicBind (NeuroSnap, Wilmington, DE, USA) [35], a deep equivariant generative model for predicting ligand-specific protein–ligand complex structures, accessed via the DynamicBind web server on NeuroSnap. Structural data for angiotensin II (PubChem CID: 172198) and angiotensin IV (PubChem CID: 123814) were provided in SDF file format for input into the simulations. Structural visualizations and images were generated and exported using PyMOL (San Francisco, CA, USA), a user-supported molecular visualization system.

### 2.2. Peptides

Angiotensin II was purchased from APExBIO Technology LLC (Houston, TX, USA). Angiotensin IV was purchased from Bachem Americas, Inc. (Torrance, CA, USA). The peptides were dissolved in deionized water at a concentration of 2 mM to prepare stock solutions and were subsequently diluted in phosphate-buffered saline (PBS) (CELLTREAT, Ayer, MA, USA) to the appropriate working concentrations for each experiment.

### 2.3. Plasmid DNAs

The plasmid DNAs, CoV2-N-WT-Hu1, CoV2-M-IRES-E, and Luc-PS9, originally developed by Doudna’s lab at the University of California, Berkeley [36], were obtained from Addgene (Watertown, MA, USA). The plasmid pcDNA3.1-SARS-S2, developed by the Fang Li lab at the University of Minnesota [37], was also obtained from Addgene.

### 2.4. Cell Culture

HEK293T cells were cultured in Dulbecco’s Modified Eagle’s Medium (DMEM; Cytiva, Marlborough, MA, USA) supplemented with 10% fetal bovine serum (FBS; Gibco, Waltham, MA, USA), 1% penicillin–streptomycin–glutamine (Gibco, Waltham, MA, USA), and 1% GlutaMAX (Gibco, Waltham, MA, USA), referred to as D10 medium. Cells were maintained in 100 mm cell culture dishes (CELLTREAT, Ayer, MA, USA) at the manufacturer-recommended seeding density, incubated in the CO_2_ incubator at 37 °C in a 5% CO_2_ environment. HEK293T-ACE2 stable cells (NR-52511), obtained from BEI Resources (NIAID, NIH, Bethesda, MD, USA), operated by American Type Culture Collection (Manassas, VA, USA), were cultured under the same conditions in 35 mm cell culture dishes using D10 medium.

For the post-infection angiotensin IV treatment, infected HEK293T cells were washed once with D10 medium and then cultured in fresh D10 medium containing 40 nM angiotensin IV for an additional 24 h in a CO_2_ incubator.

### 2.5. Immunostaining

HEK293T-ACE2 cells were stained with Alexa Fluor 647-conjugated anti-human ACE2 antibody (R&D Systems, Minneapolis, MN, USA) at a concentration of 1 µL per million cells diluted with staining buffer (Invitrogen, Carlsbad, CA, USA), following pre-incubation with 10% Normal Mouse IgG (Invitrogen, Carlsbad, CA, USA) to block Fc-receptor for non-specific binding. Viability was assessed using the Live/Dead Fixable Red Cell Stain Kit for 638 nm excitation (Invitrogen, Carlsbad, CA, USA), and cells were subsequently fixed with 2% formaldehyde (Cell Signaling Technology, Danvers, MA, USA). Immediately after fixation, samples were analyzed on a CytoFLEX flow cytometer (Beckman Coulter, Brea, CA, USA). All staining procedures were performed according to the manufacturers’ instructions.

### 2.6. SARS-CoV-2 Viral-like Particle (VLP) Production

HEK293T cells, seeded at 7 × 10^6^ per 100 mm cell culture dish pre-coated with 0.01% poly-L-lysine (Sigma, St. Louis, MO, USA), were transfected with four SARS-CoV-2 structure proteins encoding plasmid DNAs, CoV2-N-WT-Hu1, CoV2-M-IRES-E, pcDNA3.1-SARS-S2, and Luc-PS9, at a 1:1:1:1 mass ratio using polyethyleneimine (PEI) transfection reagent (Polysciences, Warrington, PA, USA). Three hours after adding the PEI–DNA mixture, the culture medium was replaced with fresh D10 medium. After 48 h, the virus-containing supernatant was harvested and filtered through a 0.45 µm sterile polyvinylidene difluoride (PVDF) membrane filter. The filtered viral supernatant was concentrated up to 10-fold using polyethylene glycol 8000 (PEG-8000: PROMEGA, Madison, WI, USA) by mixing the supernatant with PEG-8000 at a 3:1 volume ratio. The mixture was gently rotated at 60 rpm for 4 h and then centrifuged at 1600× *g* for 1 h at 4 °C. The pellet (SARS2-VLP) was resuspended in fresh D10 medium to achieve a 10-fold concentration and stored at 4 °C until use in infection assays.

### 2.7. Single-Round Infection Assay

HEK293T-ACE2 cells were seeded at 30,000 cells per well in a 96-well flat-bottom plate (CELLTREAT, Ayer, MA, USA) pre-coated with 0.01% poly-L-lysine (Sigma, St. Louis, MO, USA). Each well was then treated with 200 µL of concentrated SARS2-VLP. At 24 h post-infection, the culture medium containing VLPs was removed, and each well was washed once with 200 µL of D10 medium before replacing with fresh D10. At 48 h post-infection, luciferase activity was measured according to the manufacturer’s instructions (Promega Corporation, Madison, WI, USA) in a Molecular Devices SpectraMax Microplate Reader (Molecular Devices, San Jose, CA, USA).

### 2.8. Statistical Analysis

The error bars indicate the standard errors from multiple experiments. Statistical significance was assessed using the Wilcoxon matched-pairs signed-rank test, comparing each experimental condition to its respective control group. All statistical analyses and representations of scatter plots with bar graphs were made in GraphPad Prism 10. More specific information can be found in each figure legend.

## 3. Results

### 3.1. Molecular Docking Analysis of Angiotensin Peptides and Spike Protein Interactions

Although angiotensin II is a well-known substrate of ACE2, prompting a natural interaction with the receptor, we sought to determine whether angiotensin IV also binds to ACE2. Structural simulations of angiotensin II and IV within the spike–ACE2 complex revealed that both peptides primarily associate with the enzymatic groove of ACE2 (Figure 2A [I] and [II]) and exhibit comparable binding affinities, suggesting that their affinity for ACE2 may exceed that of the spike protein (Table 1). However, angiotensin II did not enhance spike–ACE2 interaction [31,32], raising the possibility that if angiotensin IV binds to the same ACE2 site, it may similarly have no effect on spike–ACE2 binding. Based on this, we hypothesized that angiotensin IV may also interact directly with the spike protein, particularly its trimeric form, which is the native conformation present on intact virions. Interestingly, additional protein–ligand docking simulations showed that angiotensin IV preferentially binds to the S2 domain of the closed trimeric form of the spike protein (Figure 2B [I]). However, when simulated with the open-state conformation of the SARS-CoV-2 spike ectodomain, angiotensin IV bound in close proximity to the RBD of the S1 subunit, exhibiting increased binding affinity (Figure 2B [II] and Table 1).

### 3.2. Impact of Angiotensin Peptides on SARS-CoV-2 Cell Entry Through ACE2 Receptor

To evaluate the virological relevance of the RAS in SARS-CoV-2 infectivity, we conducted a single-round infection assay using SARS-CoV-2 virus-like particles (SARS2-VLPs) engineered to express a luciferase reporter gene (Figure 3). First, we confirmed that parental HEK293T cells do not express endogenous ACE2, as determined by flow cytometry (Figure 2A). In contrast, HEK293T cells stably expressing ACE2 (HEK293T-ACE2) exhibited robust surface expression, with over 90% of the cell population positively expressing the ACE2 receptor (Figure 3A). Next, we confirmed that our single-round infection assay is ACE2-dependent by using HEK293T cells as target cells. While SARS2-VLPs failed to infect parental HEK293T cells, a significant increase in luciferase activity was observed in HEK293T-ACE2 cells following infection (Figure 3B), indicating that SARS2-VLP target cell entry occurs in an ACE2-dependent manner.

Based on the reported ELISA assay that involved pre-treatment of the spike protein with angiotensin peptides to assess the enhancement of spike protein interactions with ACE2 [31,32], we applied a similar approach in the infection assay by pre-incubating SARS2-VLP with angiotensins for one hour at 37 °C prior to infecting HEK293T-ACE2 target cells. As our preliminary data confirmed that SARS2-VLP infection is completed within 24 h post-infection (Table 2), the VLP–angiotensin peptides mixture was removed at 24 h post-infection, followed by a medium exchange after a washing step. To evaluate the dose-dependent effects of angiotensin peptides, we tested three concentrations: 40 nM, 80 nM, and 400 nM. Angiotensin II showed no effect on SARS-CoV-2 entry at any of the tested concentrations (Figure 4A). In contrast, angiotensin IV significantly increased viral infectivity at 40 nM, while higher concentrations (80 nM and 400 nM) led to a reduction in infectivity (Figure 4B), possibly due to cytotoxic or signaling effects. Notably, 40 nM angiotensin IV treatment consistently produced more than a twofold increase in luciferase activity across fifteen independent experiments, suggesting enhanced viral entry via the ACE2 receptor. However, the inhibitory effects observed at higher concentrations suggest that angiotensin IV may also exert antiviral effects that offset its entry-promoting activity. These findings highlight the unique dual role of angiotensin IV: enhancing viral entry at low concentrations predicted by increasing spike–ACE2 binding, while potentially initiating compensatory antiviral responses at higher concentrations. To further confirm that the angiotensin IV–mediated enhancement of SARS-CoV-2 target cell entry occurs specifically during the viral entry phase, rather than as a result of post-entry modulation of reporter gene expression, we treated HEK293T-ACE2 cells with 40 nM angiotensin IV at 24 h post-infection, following viral removal and medium replacement (Figure 4C). This post-infection treatment did not significantly alter luciferase activity, supporting the conclusion that angiotensin IV primarily influences the viral entry stage.

## 4. Discussion

This study demonstrates that specific downstream angiotensin peptides, particularly angiotensin IV, significantly influence how SARS-CoV-2 enters target cells via the ACE2 receptor (Figure 4B). While angiotensin II showed no measurable effect on spike–ACE2 binding, recent reports indicate that the shorter, metabolically derived peptides, angiotensin IV, significantly enhance binding affinity in ELISA-based assays [31,32]. This increased interaction may account for the transient rise in infectivity observed at lower concentrations of angiotensin IV in our single-round infection assays using VLPs (Figure 3B).

Molecular docking simulations suggest that angiotensin IV interacts not only with the enzymatic groove of ACE2 but also binds to the S2 domain of the trimeric spike protein, distinct from the spike-ACE2 binding interface (Figure 2B). Notably, angiotensin IV binds to the same domain of ACE2 as angiotensin II (Figure 2A).

Since angiotensin II does not increase the virus infection, this suggests that the ability of angiotensin IV to increase binding may be attributed to its direct interaction with the spike protein. The predicted angiotensin IV binding site on the spike protein is located within the S2 domain, spatially distinct from the RBD in the S1 subunit. This separation suggests the potential for an allosteric mechanism. In this model, angiotensin IV may induce conformational changes in the spike protein that increase its affinity for ACE2, thereby facilitating viral entry. Furthermore, in silico docking simulations also indicate that angiotensin IV binds more tightly to the open conformation of the spike protein near the RBD than to ACE2 itself (Figure 2B and Table 1), supporting the notion that angiotensin IV preferentially interacts with the open state of the spike protein to enhance spike–ACE2 engagement. Importantly, these findings raise new questions about the physiological contexts in which angiotensin IV levels are altered, particularly regarding sex and age differences. Several studies have shown that RAS activity differs between males and females. Females tend to exhibit higher ACE2 expression in specific tissues compared to males, particularly in the renal tubulointerstitial compartment of the kidney [38]. This sex difference is thought to be partly driven by estrogen, which has been reported to upregulate ACE2 expression and may contribute to the observed disparities in ACE2 levels between sexes [21,22]. This is also correlated with the fact that males tend to exhibit higher baseline levels of angiotensin II [23], while females may have relatively higher levels of angiotensin (1–7) and greater AT2R signaling, which are associated with vasoprotective and anti-inflammatory effects [24,25,39]. Current research indicates that males and females have comparable susceptibility to SARS-CoV-2 infection [40], suggesting that sex-based differences in circulating angiotensin II levels may not significantly impact viral entry, which is consistent with our in vitro findings that angiotensin II does not affect SARS-CoV-2 cellular entry mechanisms (Figure 4A). However, males are at greater risk for severe COVID-19 outcomes, including higher rates of hospitalization, ICU admission, and mortality [41,42], potentially due to elevated baseline levels of angiotensin II. While multiple biological factors likely contribute to these sex-based disparities, differences in angiotensin II concentrations may play a more prominent role in disease progression and severity rather than influencing initial viral infectivity.

While circulating levels of angiotensin II are well-documented, much less is known about angiotensin IV concentrations in circulation or within specific tissues, as it is a downstream metabolite generated through sequential enzymatic processing, including cleavage by APA. It is plausible that older adults and males, who generally experience reduced ACE2 expression and increased angiotensin II activity, may have altered conversion dynamics leading to elevated angiotensin IV levels in certain tissues, such as the lungs, heart, and kidneys, organs vulnerable to severe COVID-19. Angiotensin IV is likely present in both the CNS and peripheral tissues, as suggested by the widely distributed AT4R; however, its compartment-specific expression patterns and functional effects appear to differ between these regions [43,44,45,46]. In the CNS, angiotensin IV is generated locally and is abundantly expressed in regions such as the hippocampus, cortex, and hypothalamus, where it plays important roles in cognitive function, memory regulation, and neuroprotection [47,48]. Those regions have been reported to exhibit structural and functional alterations in COVID-19 patients, including inflammation, neuronal injury, and disrupted neurogenesis, which may contribute to cognitive impairments in COVID-19, such as brain fog [49,50]. In contrast, angiotensin IV is less abundant in plasma due to its rapid degradation and its role as a transient metabolic intermediate of angiotensin III, contributing only minimally to systemic vascular regulation [43]. Plasma angiotensin IV concentrations have been reported to vary widely, with one study indicating a median level of 39.71 ng/mL (81.6 nM) [51], while another reported a median of 8.6 pg/mL (17.7 pM) [52], suggesting substantial variability across studies [43]. Our in vitro data suggest that viral infectivity increases at 40 nM angiotensin IV, but higher concentrations beyond 80 nM reduce infectivity (Figure 4B). The median plasma concentration of angiotensin IV, based on limited two studies, is approximately 40 nM. While this estimate may lack precision, it suggests that under physiological conditions, angiotensin IV is more likely to enhance SARS-CoV-2 infection rather than inhibit it.

Studies have demonstrated that SARS-CoV-2 can directly infect human microglial cells, leading to their activation and triggering a pro-inflammatory response, including the release of IL-1β, IL-6, and TNF-α, which are associated with neuroinflammation and may underlie the neurological symptoms observed in COVID-19 [30], which closely resembles the pathogenic mechanisms seen in HIV associated neurocognitive disorders [53]. Given our findings that angiotensin IV facilitates viral entry, individuals with inherently higher local or systemic levels of angiotensin IV, particularly within the CNS, may be more susceptible to enhanced SARS-CoV-2 entry into neurons or microglial cells. This increased susceptibility could contribute to more severe neuroinvasion and may underline the neurocognitive impairments, such as brain fog, commonly observed in COVID-19 patients. A large cohort study reported that females are at significantly higher risk of developing neurological symptoms associated with COVID-19, such as fatigue, headache, brain fog, depression, and anosmia, compared to males [29]. The risk ratios for these symptoms in females range from 1.37 to 1.61, indicating a 37% to 61% higher risk than males for specific symptoms [54]. Although there is currently no direct evidence of sex-based differences in angiotensin IV levels within the CNS, such differences may contribute to the observed sex disparities in COVID-19-related neurological complications. Further clinical research is needed to quantify angiotensin peptide concentrations in both plasma and the CNS to clarify their potential role in sex-specific disease manifestations. A key limitation of our in vitro infection assays is the use of HEK293T cells, a kidney-derived cell line, which may not fully capture the cell-type-specific susceptibility influenced by angiotensin peptides. To address this, we are currently investigating the effects of angiotensin peptides on SARS-CoV-2 infection in microglial cells.

Another clinically relevant link between the RAS and SARS-CoV-2 infection is observed in individuals with obesity. Obesity is a major public health concern globally, as it disrupts multiple physiological systems, particularly immune function. It has been associated with significantly worse outcomes following SARS-CoV-2 infection, partly due to its effects on the RAS [55]. In obese individuals, circulating levels of angiotensin IV are altered, while angiotensin II levels remain relatively unchanged [51]. This suggests that changes in angiotensin IV may influence susceptibility to SARS-CoV-2 infection via the ACE2 receptor, which is also overexpressed in adipose tissue, potentially increasing viral load and persistence [55,56]. Moreover, obesity is linked to chronic activation of the RAS, promoting inflammation, vasoconstriction, and fibrosis, all of which can exacerbate lung injury when ACE2 function is impaired by viral infection [55,56,57,58]. Collectively, these factors contribute to the higher rates of hospitalization, intensive care admission, and mortality observed in individuals with obesity during COVID-19.

In summary, our study identifies angiotensin IV as a potential endogenous enhancer of SARS-CoV-2 cell entry via the ACE2 receptor, which may influence not only the acute phase of infection but also the size of the viral reservoir. These findings open new avenues for understanding how RAS imbalance, particularly involving downstream metabolites, may contribute to COVID-19 susceptibility and pathogenesis. Future research should explore angiotensin IV expression profiles across sexes, ages, and disease states, as well as the therapeutic potential of targeting angiotensin IV–related pathways to mitigate viral infectivity.

## Figures and Tables

**Figure 1 viruses-17-01014-f001:**
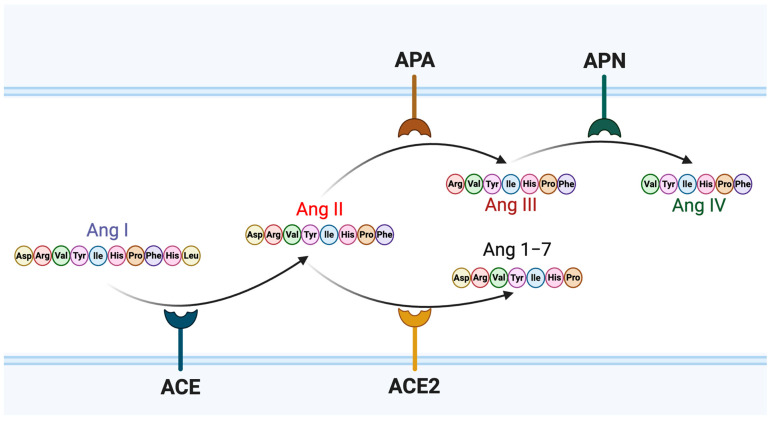
Schematic representation of amino acid sequences and metabolic pathways of angiotensin peptides. Angiotensin I (Ang I) is converted to angiotensin II (Ang II) by angiotensin-converting enzyme (ACE). Ang II can be further metabolized by ACE2 to produce angiotensin 1–7 (Ang 1–7), or by aminopeptidase A (APA) to generate angiotensin III (Ang III). Ang III is subsequently cleaved by aminopeptidase N (APN) to produce angiotensin IV (Ang IV).

**Figure 2 viruses-17-01014-f002:**
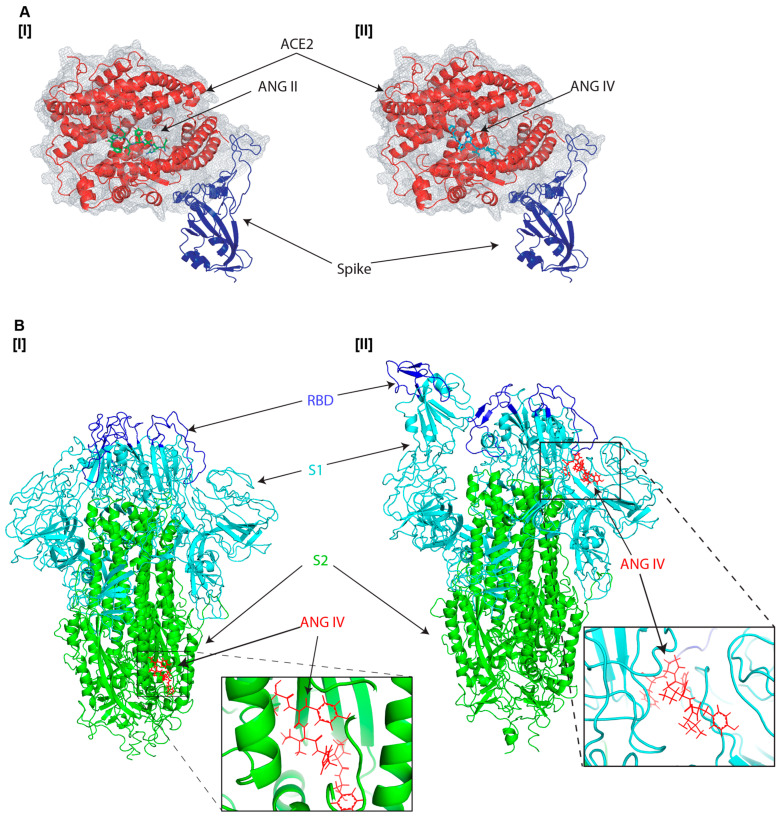
In silico docking models of angiotensin peptides with the SARS-CoV-2 spike protein and ACE2. (**A**) Predicted binding of (**I**) angiotensin II (green) and (**II**) angiotensin IV (cyan) to the SARS-CoV-2 spike receptor-binding domain (RBD, blue) in complex with ACE2 (red) based on the crystal structure (PDB: 6M0J), generated using DynamicBind. Both peptides were predicted to interact with the enzymatic groove of ACE2. Binding affinities are summarized in Table 1. (**B**) Simulated binding of angiotensin IV (red) to the SARS-CoV-2 spike trimer in its (**I**) closed (PDB: 6VXX) or (**II**) open (PDB: 6VYB) conformations. In these models, the RBD is shown in blue, the S1 domain in cyan, and the S2 domain in green.

**Figure 3 viruses-17-01014-f003:**
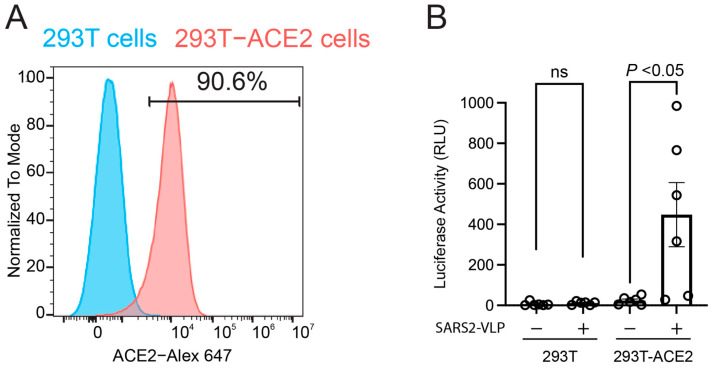
Single-round infection assay with SARS-CoV-2 VLP in 293T-ACE2 cells. (**A**) Flow cytometry analysis confirming ACE2 receptor expression in HEK293T-ACE2 cells used as target cells in the infection assay. HEK293T-ACE2 and parental HEK293T cells were stained with an Alexa Fluor 647-conjugated anti-ACE2 antibody. Parental HEK293T cells showed no detectable ACE2 expression, whereas HEK293T-ACE2 cells exhibited robust surface ACE2 expression. (**B**) Viral infectivity was assessed in HEK293T-ACE2 cells and parental HEK293T cells using a luciferase reporter assay to confirm that SARS-CoV-2 VLP infection is ACE2-dependent. Error bars indicate standard error from six samples and statistical significance was calculated by the Wilcoxon matched-pairs signed-rank test. Each circle represents an individual trial, and the label “ns” indicates that the results of that trial was not statistically significant.

**Figure 4 viruses-17-01014-f004:**
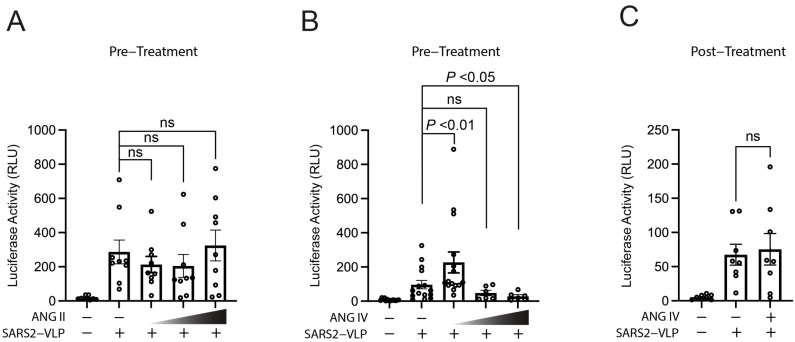
Effects of angiotensin II and IV on the viral entry step of SARS-CoV-2 infection. SARS2-VLPs were pre-treated with angiotensin peptides at concentrations of 40 nM, 80 nM, and 400 nM prior to infection of HEK293T-ACE2 cells. (**A**) Angiotensin II treatment and (**B**) angiotensin IV treatment were assessed for their effects on viral entry. (**C**) To examine the post-entry effects, HEK293T-ACE2 cells were first infected with SARS2-VLPs and incubated for 24 h. After removing residual virus by washing, cells were treated with 40 nM angiotensin IV for an additional 24 h before measuring luciferase activity. Viral entry efficiency was quantified 48 h post-infection using a luciferase reporter assay. (**A**) For angiotensin II, 9 independent experiments were conducted at each concentration. (**B**) The 40 nM angiotensin IV group included 15 independent experiments, while the 80 nM and 400 nM groups included 6 independent experiments each. (**C**) A total of 8 independent experiments were performed for the post-entry 40 nM angiotensin IV treatment. Statistical significance was evaluated using the Wilcoxon matched-pairs signed-rank test: 9 matched pairs for angiotensin II compared to untreated samples and 15 matched pairs for the 40 nM angiotensin IV pre-treatment group, 6 matched pairs for the 80 nM and 400 nM groups or 8 matched pairs for angiotensin IV post-treatment groups, respectively, compared to untreated samples. Error bars represent the standard error of the mean. Each circle represents an individual trial, and the label “ns” indicates that the results of that trial was not statistically significant.

**Table 1 viruses-17-01014-t001:** DynamicBind Metrics.

Protein	Ligand	cLDDT	Binding Affinity
SARS-CoV-2 spike receptor-binding domain bound with ACE2 (PDB: 6M0J)	Angiotensin II	0.5082692	5.285819
SARS-CoV-2 spike receptor-binding domain bound with ACE2 (PDB: 6M0J)	Angiotensin IV	0.6030191	4.9195795
SARS-CoV-2 spike glycoprotein (closed state) (PDB: 6VXX)	Angiotensin II	0.47392076	3.6205282
SARS-CoV-2 spike ectodomain structure (open state) (PDB: 6VYB)	Angiotensin II	0.34993827	6.679925

Contact Local Distance Difference Test (cLDDT): The accuracy of the predicted binding complex; a higher match to the true binding, with 1.0 signifying near-native contacts. Binding Affinity: the strength of the interaction between the protein and ligand; a higher value indicates a stronger predicted affinity.

**Table 2 viruses-17-01014-t002:** Luciferase activity in different virus exposure times.

Sample	Luc (RLU.)	Virus Exposure Time
No VLP	123.46	24 h
	218.49	48 h
SARS2-VLP	2192.15	24 h
	1642.48	48 h

Luciferase assay was performed 48 h post-infection. For the virus exposure time of twenty-four hours, the VLP-containing medium was washed out with fresh D10 medium and incubated additionally for twenty-four hours, while the other samples were incubated with VLP-containing medium for forty-eight hours.

## Data Availability

Data is contained within the article.

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
