# Peer review of "The Renin–Angiotensin System Modulates SARS-CoV-2 Entry via ACE2 Receptor"

_viruses, 2025, doi:10.3390/v17071014_

Round 1
Reviewer 1 Report
Comments and Suggestions for Authors
Dear Dr.,
Title: The Renin–Angiotensin System Modulates SARS-CoV-2 Entry via ACE2 Receptor
Manuscript ID: viruses-3748844
Overall comments: Gagliardi et al. described in this manuscript; the novel link of RAS-derived peptides and SARS-CoV-2 infectivity for the progression of COVID-19 pathophysiology and possible therapeutic strategies. The limitation of this article is that, it has numerous typographical errors. The overall manuscript is good, and it can help those working in this field of research.
Specific comments:
- The author’s number and symbol must be superscript in the author list.
- In abstract: expansion of the abbreviation should be mentioned in the first appearance location.
- Keywords section: no need for abbreviated terms.
- In the introduction, figure 1 should be black and white, and the resolution needs to be improved.
- What is the rationale for the selection of HEK293T cells in this study? There are more valuable in vitro cell culture models e., Calu-3 cell line used for the SARS-CoV-2 virus, and also A549 cells.
- In 2.3. Cell Culture section: multiple small paragraphs can be avoided.
- The section ‘2.7. In Silico Docking Simulation’ can be placed in the location of section 2.1.
- Section 2.8 must be expanded with hypothesis testing and P-value indication, and software used, etc.
- Letter figure 2, no need to highlight with red colour. The resolution of figures 2 and 3 needs to improve.
- Line 245 highlighted word needs to be rectified. Similarly, all other highlighted text needs to be rectified.
- Table 1 text information was expressed a congested manner. Need to make concise and clear manner.
- Figure 4 A[I] and [II]: docking binding packed need to be enlarged, and the resolution needs to improve. Docking results can bring the beginning of the result section, as per the methods section suggestion.
- The discussion is too vague; it can be made more concise and clear. The limitation of this needs to be expressed before the summary statement.
- Recent relevant references can be incorporated as follows:
Mahdi M, Kiarie IW, Mótyán JA, Hoffka G, Al-Muffti AS, Tóth A, TÅ‘zsér J. Receptor Binding for the Entry Mechanisms of SARS-CoV-2: Insights from the Original Strain and Emerging Variants. Viruses. 2025 May 10;17(5):691.
Muthuraman A, Kaur P. Renin-Angiotensin-Aldosterone System: A Current Drug Target for the Management of Neuropathic Pain. Curr Drug Targets. 2016;17(2):178-95. doi: 10.2174/1389450116666150825115658. PMID: 26302799.
Xing L, Liu Z, Wang X, Liu Q, Xu W, Mao Q, Zhang X, Hao A, Xia S, Liu Z, Sun L. Early fusion intermediate of ACE2-using coronavirus spike acting as an antiviral target. Cell. 2025 Jan 30.
hoon Lee J, Sergi C, Kast RE, Kanwar BA, Bourbeau J, Oh S, Sohn MG, Lee CJ, Coleman MD. Basic implications on three pathways associated with SARS-CoV-2. Biomedical Journal. 2024 Jul 14:100766.
Muthuraman A, Kaur J. Antimicrobial nanostructures for neurodegenerative infections: present and future perspectives. In Nanostructures for Antimicrobial Therapy 2017 Jan 1 (pp. 139-167).
Minor comments
- Reference updates are required.
- Numerous typographical errors were observed; it’s must be rectified.
*****
Author Response
Dear Reviewer 1
Thank you so much for your time in reviewing our manuscript and providing us with valuable feedback.
Overall comments: Gagliardi et al. described in this manuscript; the novel link of RAS-derived peptides and SARS-CoV-2 infectivity for the progression of COVID-19 pathophysiology and possible therapeutic strategies. The limitation of this article is that, it has numerous typographical errors. The overall manuscript is good, and it can help those working in this field of research.
Response to the Reviewer's Overall Comments: We have revised it accordingly. Each of your comments is addressed in detail below.
Reviewers Comment 1: The author’s number and symbol must be superscript in the author list.
Response to the Reviewer's Comment 1: Thank you for pointing this out to us. Though this format was automatically generated in the Viruses submission system, which was not under our control for the detailed format, we superscripted the authors' numbers and symbols accordingly in the uploaded Word file.
Reviewers Comment 2: In abstract: expansion of the abbreviation should be mentioned in the first appearance location.
Response to the Reviewer's Comment 2: Thank you for your feedback. While the instructions for authors in the Viruses journal indicate the abstract should be a total of about 200 words maximum, we described the full name of each abbreviation in the abstract accordingly, and the current word count is 217, which we hope is within about 200 words.
Reviewers Comment 3: Keywords section: no need for abbreviated terms.
Response to the Reviewer's Comment 3: All abbreviated terms have been pulled from the keywords section and replaced with the appropriate expansions.
Reviewers Comment 4: In the introduction, figure 1 should be black and white, and the resolution needs to be improved.
Response to the Reviewer's Comment 4: It seems that the PDF file compressed the resolution of some figures, we replaced Figure 1 in the Word file with the required resolution accordingly. We believe the color differences in each amino acid enhance the visual understanding of which residue was cleaved off in each metabolic process. In addition, Viruses Journal is an online journal that does not charge color costs for publication. While we truly respect and appreciate the reviewer's comments, we have decided to keep the color of the figure for these reasons. We appreciate your understanding.
Reviewers Comment 5: What is the rationale for the selection of HEK293T cells in this study? There are more valuable in vitro cell culture models e., Calu-3 cell line used for the SARS-CoV-2 virus, and also A549 cells.
Response to the Reviewer's Comment 5: HEK293T cells were used since they do not express any receptors necessary for the SARS-CoV-2 cellular entry process. Therefore, HEK293T cells are not susceptible to being infected with SARS-CoV-2, which we confirmed in Figure 3B. Since we specifically focus on the effect of angiotensin peptides on SARS-CoV-2 entry through the ACE2 receptor, HEK293T-ACE2 cells are the ideal cellular resource. While Calu-3 cells express ACE2 (Ren, Glende et al. 2006), this cell line may also express NRP1, another receptor responsible for SARS-CoV-2 infection, due to their origin from human lung carcinoma cell lines, in which NRP-1 expression has been demonstrated in Calu-1, the same lung carcinoma cell line. The same reason to A549 cells. A549 cells express NRP-1(Jimenez-Hernandez, Vazquez-Santillan et al. 2018), but A549 cells do not express the ACE2 receptor (Chang, Parsi et al. 2022).
Reviewers Comment 6: In 2.3. Cell Culture section: multiple small paragraphs can be avoided.
Response to the Reviewer's Comment 6: Though we break the line of the cell culture procedure in the infection assay to distinguish it from the regular passage procedure of cell culture, we do not create any small paragraphs in this subparagraph. We believe the regular cell culture procedure should be separated from the cell culture procedures in the infection assay by breaking the sentence line. In response to the next comment, this subparagraph is currently Section 2.4.
Reviewers Comment 7: The section ‘2.7. In Silico Docking Simulation’ can be placed in the location of section 2.1.
Response to the Reviewer's Comment 7: According to the reviewer's comment, we replaced section 2.7, In Silico Docking Simulation, with subsection 2.1, and the following subparagraph numbers were adjusted in the Materials and Methods section.
Reviewers Comment 8: Section 2.8 must be expanded with hypothesis testing and P-value indication, software used, etc.
Response to the Reviewer's Comment 8: According to the reviewer's comment, we have expanded upon this section in lines 185 - 189 on page 5.
Reviewers Comment 9: Letter figure 2, no need to highlight with red colour. The resolution of figures 2 and 3 needs to improve.
Response to the Reviewer's Comment 9: According to the reviewer's comments, we replaced Figures 2 and 3 with the required resolution. The text color to indicate the responsible figures in the manuscript was changed to black.
Reviewers Comment 10: Line 245 highlighted word needs to be rectified. Similarly, all other highlighted text needs to be rectified.
Response to the Reviewer's Comment 10: Thank you for pointing it out. We fixed it accordingly.
Reviewers Comment 11: Table 1 text information was expressed a congested manner. Need to make concise and clear manner.
Response to the Reviewer's Comment 11: According to the reviewer's comment, we revised and condensed the information in Table 1.
Reviewers Comment 12: Figure 4 A[I] and [II]: docking binding packed need to be enlarged, and the resolution needs to improve. Docking results can bring the beginning of the result section, as per the methods section suggestion.
Response to the Reviewer's Comment 12: According to the reviewers' comments, we revised Figure 4 and moved it to Figure 2, adjusting the subsequent figure numbers accordingly. Additionally, the subparagraph on Molecular Docking Analysis of Angiotensin Peptides and Spike Protein Interactions has been moved to the first position as 3.1, while the original subparagraph 3.1 has been relocated to 3.2 in the results section.
Reviewers Comment 13: The discussion is too vague; it can be made more concise and clear. The limitation of this needs to be expressed before the summary statement.
Response to the Reviewer's Comment 13: According to the reviewer's comment, we deleted the description of the possibility of cytotoxic effects by angiotensin peptides in HEK293T cells because it may not be a major issue. In addition, we described the key limitation in our experiment in line 373 - 377 on page 11.
Reviewers Comment 14: Recent relevant references can be incorporated as follows:
Mahdi M, Kiarie IW, Mótyán JA, Hoffka G, Al-Muffti AS, Tóth A, TÅ‘zsér J. Receptor Binding for the Entry Mechanisms of SARS-CoV-2: Insights from the Original Strain and Emerging Variants. Viruses. 2025 May 10;17(5):691.
Muthuraman A, Kaur P. Renin-Angiotensin-Aldosterone System: A Current Drug Target for the Management of Neuropathic Pain. Curr Drug Targets. 2016;17(2):178-95. doi: 10.2174/1389450116666150825115658. PMID: 26302799.
Xing L, Liu Z, Wang X, Liu Q, Xu W, Mao Q, Zhang X, Hao A, Xia S, Liu Z, Sun L. Early fusion intermediate of ACE2-using coronavirus spike acting as an antiviral target. Cell. 2025 Jan 30.
Hoon Lee J, Sergi C, Kast RE, Kanwar BA, Bourbeau J, Oh S, Sohn MG, Lee CJ, Coleman MD. Basic implications on three pathways associated with SARS-CoV-2. Biomedical Journal. 2024 Jul 14:100766.
Muthuraman A, Kaur J. Antimicrobial nanostructures for neurodegenerative infections: present and future perspectives. In Nanostructures for Antimicrobial Therapy 2017 Jan 1 (pp. 139-167).
Response to the Reviewer's Comment 14: We have incorporated relevant references that were deemed necessary in line 41 on page 1, line 45 and line 48 on page 2, and line 319 on page 10. The last article the reviewer suggested to refer to is not relevant to our paper because it discusses nanostructures and their involvement in changes to the drug delivery system, which we do not address in the manuscript. Therefore, we decided not to reference this paper.
Minor comments
Reviewers Comment 15: Reference updates are required.
Response to the Reviewer's Comment 15: References have been updated.
Reviewers Comment 16: Numerous typographical errors were observed; it’s must be rectified.
Response to the Reviewer's Comment 16: We read through the manuscript multiple times to check for typos and fixed them as much as we could.
Reviewer 2 Report
Comments and Suggestions for Authors
In the manuscript presented by Gagliardi et al., the importance of the renin-angiotensin system as a modulator of SARS-CoV-2 entry via the ACE2 receptor is raised. This study is interesting and could be very useful in gaining a better understanding of the differences in the development of clinical complications in patients who develop SARS-CoV-2 disease. The methodology used is sufficient considering the objective of the study, and the results support the discussion. However, I have the following comments.
I. Comments:
1. Improve the wording of the study objective.
2. Obesity (a major public health problem in many countries) influences the renin-angiotensin system, in addition to generating an increase in the inflammatory response (chronic subclinical inflammation). Discuss this point in the context of SARS-CoV-2.
3. Improve the resolution of the figures.
4. In the abstract, it would be helpful if the authors defined the abbreviations.
5. What would be the scope of the study, especially considering the high prevalence of chronic diseases in adults?
Author Response
Dear Reviewer 2
Thank you so much for your time in reviewing our manuscript and for providing us with valuable feedback. We have responded to each comment below.
Reviewers Comment 1: Improve the wording of the study objective.
Response to the Reviewer's Comment 1: According to the reviewer's comment, we revised the objective of this study in the introduction section in lines 103-106 on page 3.
Reviewers Comment 2: Obesity (a major public health problem in many countries) influences the renin-angiotensin system, in addition to generating an increase in the inflammatory response (chronic subclinical inflammation). Discuss this point in the context of SARS-CoV-2.
Response to the Reviewer's Comment 2: According to the reviewer's comment, we also discussed the RAS contribution to SARS-CoV-2 infection in obesity in lines 378-390 on page 11.
Reviewers Comment 3:. Improve the resolution of the figures.
Response to the Reviewer's Comment 3: It seems that the PDF file compressed the resolution of some figures, we replaced all figures in the Word file with the required resolution accordingly.
Reviewers Comment 4: In the abstract, it would be helpful if the authors defined abbreviations.
Response to the Reviewer's Comment 4: Thank you for your feedback. While the instructions for authors in the Viruses journal indicate the abstract should be a total of about 200 words maximum, we described the full name of each abbreviation in the abstract accordingly, and the current word count is 217, which we hope is within about 200 words.
Reviewers Comment 5: What would be the scope of the study, especially considering the high prevalence of chronic diseases in adults?
Response to the Reviewer's Comment 5: The aim of this study is to investigate how the RAS influences SARS-CoV-2 infection, potentially contributing to sex-specific differences in the prevalence of chronic disease outcomes by modulating the initial viral reservoir size. To clarify this objective, we have revised the relevant sentences in the Introduction in lines 108–111, page 3 and Discussion in lines 391–393, page 11.
Round 2
Reviewer 1 Report
Comments and Suggestions for Authors
None.